# Toward Avoiding Misalignment: Dimensional Synthesis of Task-Oriented Upper-Limb Hybrid Exoskeleton

**Sakshi Gupta** **, Anupam Agrawal and Ekta Singla \***

Mechanical Engineering Department, Indian Institute of Technology Ropar, Rupnagar 140001, India:
2016mez0024@iitrpr.ac.in (S.G.); anupam@iitrpr.ac.in (A.A.)
\* Correspondence: ekta@iitrpr.ac.in

**Abstract:** One of the primary reasons for wearable exoskeleton rejection is user discomfort caused by misalignment between the coupled system, i.e., the human limb and the exoskeleton. The article focuses primarily on the solution strategies for misalignment issues. The purpose of this work is to facilitate rehabilitative exercise-based exoskeletons for neurological and muscular disorder patients, which can aid a user in following the appropriate natural trajectory with the least amount of misalignment. A double four-bar planar configuration is used for this purpose. The paper proposes a methodology for developing an optimum task-oriented upper-limb hybrid exoskeleton with low active degrees-of-freedom (dof) that enables users to attain desired task space locations (TSLs) while maintaining an acceptable range of kinematic performance. Additionally, the study examines the influence of an extra restriction placed at the elbow motion and the compatibility of connected systems. The findings and discussion indicate the usefulness of the proposed concept for upper-limb rehabilitation.

**Keywords:** dimensional synthesis; task-oriented configuration; joint-misalignment; upper-limb exoskeleton

## 1. Introduction

Patients with neurological and muscular diseases cannot move their limbs due to poor sensory and motor skills [1]. According to the report, the only treatment available for disabled people is *repetitious physiotherapy*. Experts recommend that regular movement-based training may assist in reactivating impaired sensory function and increasing their efficiency and dependability in performing daily tasks. Robotic therapy is considered to be well suited for the purpose of improving patient recovery rates [2]. Several robotic treatment devices have been developed to help with upper-extremity rehabilitation [3–7]. Only a few of these prototypes have been marketed as a result of their limitations, as detailed below.

In recent studies [8–11], it has been recommended that rehabilitation programs target specific muscles and ligaments with more intense and regulated activities. Rather than recreating a whole human workspace, splitting the workspace is the best option. Rehabilitation firms are increasingly using rehab devices for upper extremity recovery because they can execute a greater number of therapeutically helpful movements in a smaller area. Task-based studies have been employed by researchers to develop upper-limb robotic rehabilitation devices that concentrate largely on activities in daily living (ADL), even though the range of motion (ROM) is the first step before gaining independence in ADL [12,13]. Mismatched rotational axes, high power to weight ratio, kinematic compatibility difficulties, and non-repetitive inverse solution may all result from serial connections in ADL-based manipulators with several degrees-of-freedom (dof) [6,14,15]. On the other hand, researchers seek medically relevant motions with greater manipulability and positional reachability. The field lacks the contributions in task-oriented design for synthesizing robotic assistance with the lowest possible active dof. Second, serially linked connections are usually used to achieve high manipulability, but parallel manipulators are

used to achieve greater positional reachability [16–20]. This concept inspired the use of hybrid configurations in this work. The shifting instantaneous center gives the flexibility required to address misalignment and kinematic compatibility. Thus, an adequate hybrid configuration for simulating natural human motion is required.

This research focuses on a novel strategy utilizing a hybrid configuration to construct a 2-dof task-based rehabilitation device for the recovery of shoulder and elbow flexion/extension movement while preventing joint misalignment and enhancing user comfort as well as avoiding a large number of active dof. This is accomplished by incorporating the characteristics of a double-four bar mechanism and by performing dimensional synthesis.

Major aspects addressed in this paper in order to synthesize the architecture for rehabilitation aid proposed are as follows:

- Task-based synthesis is used to design a customized upper-limb rehabilitation device with a minimal number of active dof capable of acquiring therapeutically desirable movements (ROM exercises).
- Designing and evaluating an optimal double four-bar configuration to mimic natural human motion and minimize misalignment and singularity concerns is offered.

## 2. Kinematic Compatibility

A major limitation of the usability of exoskeleton is its kinematic incompatibility with the wearer. This occurs due to mismatches of the centers of rotation of the wearers joints with those of the corresponding exoskeleton joints. Human–robot compatibility majorly consists of the following steps:

- Match number of dof between the robotic system and the wearer [21];
- Minimize variation of instantaneous center of rotation between the robot and wearer [21];
- Identify coupling relationship between robot and the wearer [6,22].

Elbow joint rotation is a multi-axis joint rotation, i.e., the instantaneous center of rotation of elbow axis varies with the elbow flexion–extension movement. It is reported that, normally, 2.5 mm × 7.8 mm is the cross-sectional area of the instantaneous center of elbow joint at lateral view (sagittal plane) [23,24]. Elvire et al. [24] reported in their paper that the center of rotation of the elbow is $7 \pm 14$ mm at distal, $4 \pm 9$ mm at lateral and $4 \pm 10$ mm at the anterior to medial epicondyle. Figure 1 shows the varying instantaneous center position with respect to the elbow flexion–extension motion. During the motion, the humerus is fixed and the ulna moves with respect to its varying instantaneous center.

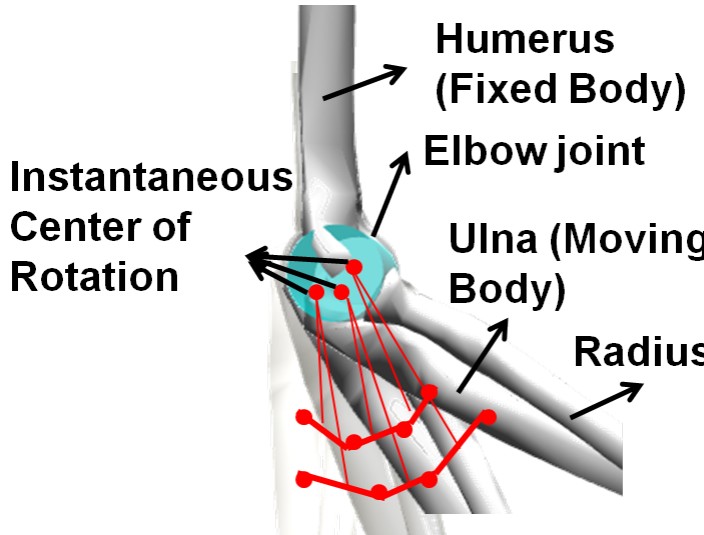

**Figure 1.** Representation of varying instantaneous center with elbow flexion/extension movement.

Figure 2 depicts the integration of a conceptual planar exoskeleton with the wearer. It represents the generation of residual forces in 2 active dofs anthropometric exoskeleton during the elbow flexion/extension motion. Figure 2 represents the effects of variation of instantaneous center on the harness position. The exoskeleton end-effector ($P$) tries to compensate with the created linear ($d_x$) and angular ($d_y$) displacement, but the harness and less flexibility in design is restricted for it. Thus the mechanism which has a single-axis revolute joint at the biological joint generates residual forces at the end-effector and may become uncomfortable to the wearer.

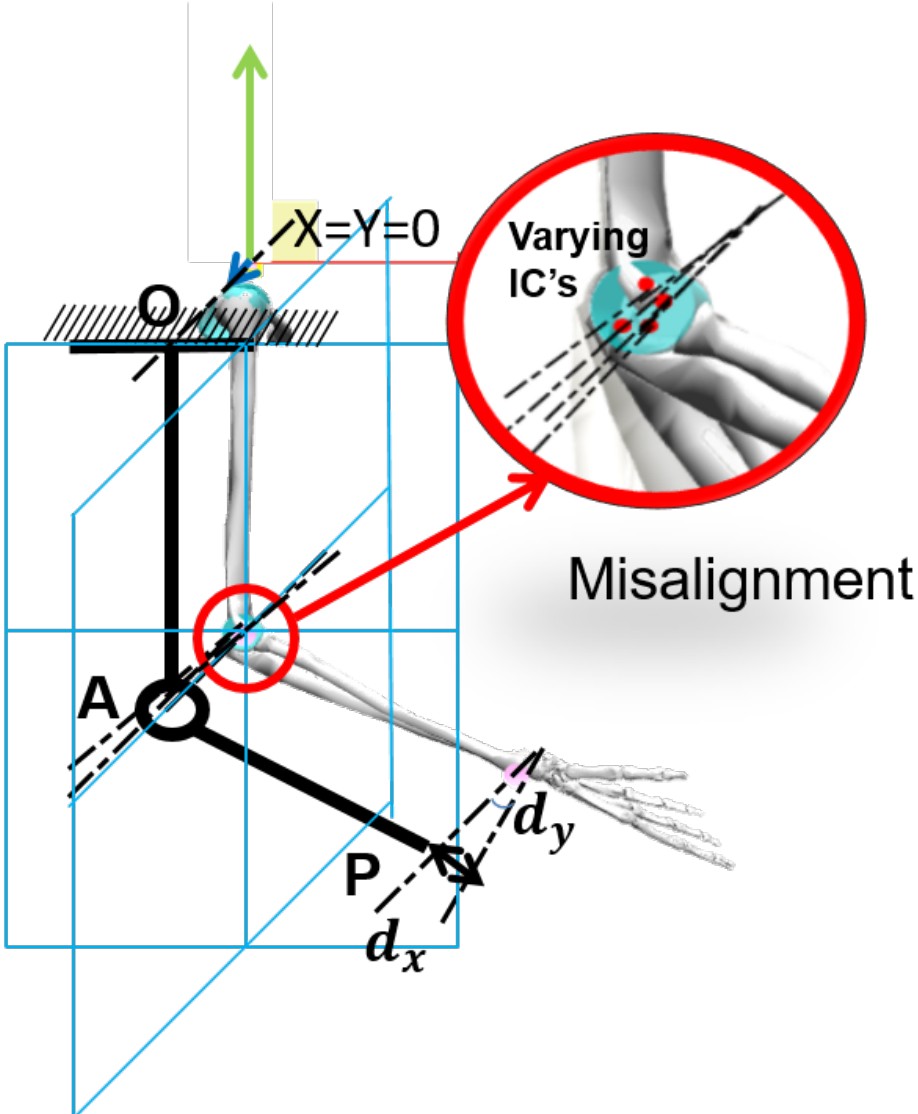

**Figure 2.** Schematic representation of effect of misalignment between exoskeleton and human limb during elbow flexion/extension motion, depicts that axis of rotation varies and creates linear ($d_x$) and angular displacement ($d_y$).

Figure 2 illustrates a closed loop formed by an exoskeleton and a human limb. It is important to compute the number of dof for a closed-loop chain, and that can be represented as Equation (1).

$$F = \sum_{i=1}^{m} f_i - dl \tag{1}$$

Here, $F$ represents the dof of the multi-loop chain (closed loop formed by exoskeleton and human arm), $f_i$ denotes the $i$th joint, $m$ indicates number of joints, $l$ denotes the number

of separate loops, and $d$ represents the function of motion space in a closed loop (for planar $d = 3$ and for spacial $d = 6$). Equation (1) can be rearranged as

$$F = f_k + f_{uk} - dl = \sum_{i=1}^{j} f_i + \sum_{s=j+1}^{m} f_s - dl \tag{2}$$

Here, $f_k$ and $f_{uk}$ represent the total known dof (active dof) and total unknown dof (passive dof), respectively. Thus, for 2-dof planar exoskeleton as shown in Figure 2, the closed loop consists of dof of the multi-loop chain, $F$ is 2; total known dof (active dof) $f_k$ is 4; function of motion space $d$ is 3; and the number of separate loops $l$ is 1. Therefore, the total unknown dof (passive dof) $f_{uk}$ is computed by Equation (2) as

$$f_{uk} = F - f_k + dl = 2 - 4 + (3*1) = 2 - 4 + 3 = 1 \tag{3}$$

This proposed passive actuator can be revolute or prismatic. However, based upon the desired demand, i.e., the actuator should be capable of compensating linear and angular displacement which are generated during the misalignment compensation as shown in Figure 2. A single revolute passive joint may create an issue. As shown in Figure 3, $A$ is attached to the revolute passive joint and may hurt the human. However, it is discussed earlier that as the biological elbow has a multi-axis joint, the exoskeleton should also have multi-joint movement. Therefore, the closed-loop concepts are introduced in the exoskeleton mechanism and consider its characteristics during the exoskeleton configuration selection. Equation (3) provides two active dof. The work selected a hybrid configuration with 2-dof, i.e., a four-bar loop connected to another four-bar loop with a common bar. Figure 4 shows the two separate four bar loops and their combination. Point, $P$ is the end effector of the exoskeleton and is capable of moving along with varying ICs (instantaneous centers). The two conditions of the attachment of a double-four bar exoskeleton with the human limb, i.e., non-anthropomorphic (end-effector) and anthropomorphic types, are demonstrated in the paper.

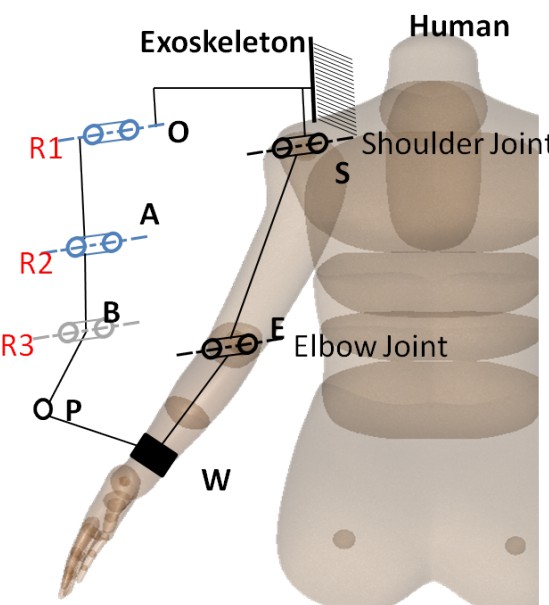

**Figure 3.** Compatibility between humans and robots is illustrated by the additional revolute joint that is attached at point $A$.

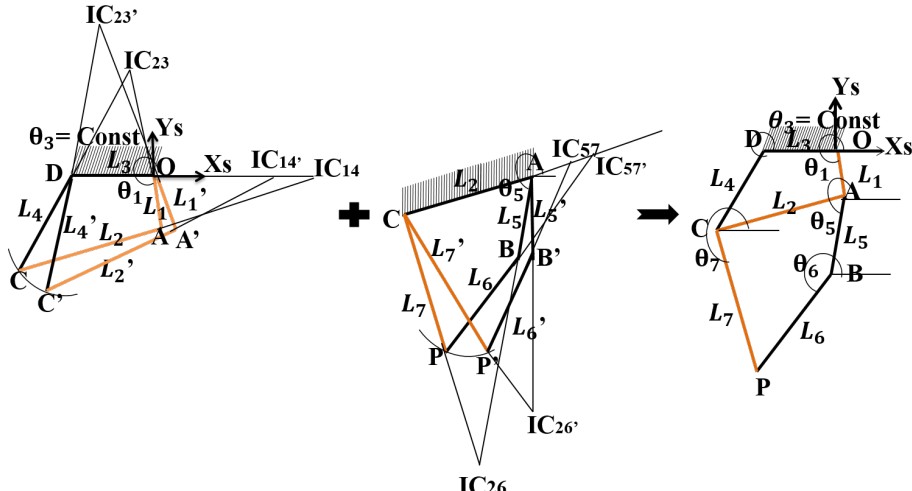

**Figure 4.** Representation of varying instantaneous center of the two separate four-bar mechanism and their combination.

## 3. Methodology

In order to design a task-oriented upper-limb rehabilitation exoskeleton that avoids misalignment and increases wearer comfort, a four-step strategy is proposed.

### 3.1. Measurement of Natural Human Motion Trajectory

To determine the natural trajectory of human motion, first choose a rehabilitation activity and then compute the associated real-time motion data. Numerous strategies, both traditional and nontraditional, can be employed to accomplish this. The common strategies for extracting natural human motion data are based upon marker-less and marker-based methods. However, for disabled patients, neither strategy is successful. The study therefore provides a simulation environment to obtain the normal human mobility for disabled patients based on their anthropometric data, as mentioned in Section 3.1.2.

#### 3.1.1. Selection of Rehabilitation Exercises

To begin, standard rehabilitation exercises were chosen based on expert suggestions (*courtesy: Stroke Rehabilitation Center—Indian Spinal Injury Center, Delhi, India*) and accessible anthropometric exercise-based data (see Appendix A) in rehabilitation centre (Indian Spinal Injury Center, Delhi, India). Planar exercises were chosen as demonstrations because of their simplicity and ability to validate conceptual design assumptions in a standardized manner. Appendix A provides a list of upper-limb exercises. The article shows the effects of planar exercise, such as lifting the right hand. Other recommended planar exercises can be carried out in a similar manner.

#### 3.1.2. Human Motion Data Collection

Quantitative gesture analysis is an important tool for measuring normal and unusual patterns of motion and has been found to be beneficial for obtaining the kinematic modeling of the upper limb. Normally, the data consist of relative information among body segments, say, positions and orientations of the prominent motion steps. The task space locations are recorded for different segments of an upper limb while performing the recommended exercises.

It is anticipated that Open-Sim would be able to replicate movement rapidly and correctly, even when it comes to patients with physical disabilities [25]. The investigation included simulations of dynamic movement, and neuromuscular coordination and physical performance were examined. Toward this, the study collected participants' natural human motion data sets, created utilizing scaled Open-Sim upper extremity musculoskeletal models. The program anticipates natural motion coordinate data by using motor control models,

such as kinematic adaptations of human gestures at various phases. The step-by-step procedure to obtain the subject-specific scaled musculoskeletal model is represented below.

1. Initialize D-H parameters for static pose (link length, joint offset, twist angle, and joint angle) by utilizing anthropometric subject data.
2. Define D-H parameter in RoboAnalyzer software.
3. Set time duration and number of steps ($t$, $n$).
4. Compute forward kinematics.
5. Extract data points for static pose.
6. Import in Open-Sim as a markers data for static pose.
7. Compute scale factor using Open-Sim GUI.
8. Run the simulation.

A large data set was obtained. Only a few sample coordinate points are shown in Table 1, which represents the transformed co-ordinate measurement of shoulder, elbow and wrist position, where the shoulder joint is considered a local co-ordinate frame.

**Table 1.** Transformed co-ordinates of shoulder, elbow and wrist joint with respect to shoulder as reference frame for the selected task: lifting right hand. All dimensions are in 'meters'.

| $X_{shoulder}$ | $Y_{shoulder}$ | $X_{elbow}$ | $Y_{elbow}$ | $X_{wrist}$ | $Y_{wrist}$ |
| --- | --- | --- | --- | --- | --- |
| 0 | 0 | −0.293 | −0.0432 | −0.530 | 0.023 |
| 0 | 0 | −0.254 | 0.109 | −0.470 | 0.240 |
| 0 | 0 | −0.163 | 0.198 | −0.333 | 0.397 |
| 0 | 0 | 0.144 | 0.301 | 0.186 | 0.575 |
| 0 | 0 | 0.166 | 0.298 | 0.225 | 0.571 |

### 3.2. Kinematic Analysis of Double-Four Bar Configuration

In this section, kinematic analysis of human robot exoskeleton is defined for double-four bar connected in series. The configuration's reference frame is aligned to *X*-axis and assuming its lengths named $L_1$ to $L_7$ as shown in Figure 5. The configuration has two active joints, $\theta_1$ and $\theta_5$. All the joint angles are considered in anti-clockwise direction from *X*-axis, marked with $\theta_1$ to $\theta_7$. All the other joints $\theta_2$, $\theta_4$, $\theta_6$, and $\theta_7$ are inactive joints and can be expressed in terms of $\theta_1$ and $\theta_5$.

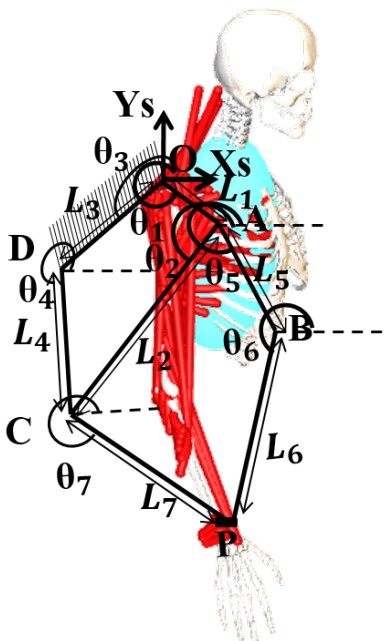

**Figure 5.** Schematic pictorial representation of double-four bar configuration coupled with human limb.

The closed-loop equations of the system are given in Equation (4),

$$
\begin{aligned}
F_1 &= \quad L_1 Cos\theta_1 + L_2 Cos\theta_2 - L_3 Cos\theta_3 - L_4 Cos\theta_4 \quad = 0, \\
F_2 &= \quad L_1 Sin\theta_1 + L_2 Sin\theta_2 - L_3 Sin\theta_3 - L_4 Sin\theta_4 \quad = 0, \\
F_3 &= \quad L_5 Cos\theta_5 + L_6 Cos\theta_6 - L_2 Cos\theta_2 - L_7 Cos\theta_7 \quad = 0, \\
F_4 &= \quad L_5 Sin\theta_5 + L_6 Sin\theta_6 - L_2 Sin\theta_2 - L_7 Sin\theta_7 \quad = 0.
\end{aligned}
\tag{4}
$$

Consider elbow point $C$ as a set of elbow-space locations (ESLs), which is computed in Equation (5) as

$$
\begin{aligned}
C_x &= \quad L_1 Cos\theta_1 + L_2 Cos\theta_2, \\
C_y &= \quad L_1 Sin\theta_1 + L_2 Sin\theta_2.
\end{aligned}
\tag{5}
$$

The location of the end-effector, considered at point $P$, is computed in Equation (6), through the path OACP.

$$
\begin{aligned}
P_x &= \quad L_1 Cos\theta_1 + L_2 Cos\theta_2 + L_7 Cos\theta_7, \\
P_y &= \quad L_1 Sin\theta_1 + L_2 Sin\theta_2 + L_7 Sin\theta_7.
\end{aligned}
\tag{6}
$$

Equation (7) illustrates the Jacobian obtained using these point locations of the double four-bar mechanism.

$$
\begin{aligned}
J &= \begin{bmatrix} \frac{\delta P_x}{\delta \theta_1} & \frac{\delta P_x}{\delta \theta_5} \\ \frac{\delta P_y}{\delta \theta_1} & \frac{\delta P_y}{\delta \theta_5} \end{bmatrix} \\
&= \begin{bmatrix} -L_1 Sin\theta_1 - L_2 Sin\theta_2 \frac{\delta \theta_2}{\delta \theta_1} & -L_7 Sin\theta_7 \frac{\delta \theta_7}{\delta \theta_5} \\ L_1 Cos\theta_1 + L_2 Cos\theta_2 \frac{\delta \theta_2}{\delta \theta_1} & L_7 Cos\theta_7 \frac{\delta \theta_7}{\delta \theta_5} \end{bmatrix}.
\end{aligned}
\tag{7}
$$

From Equations (4) and (5), the values of $\frac{\delta \theta_2}{\delta \theta_1}$ and $\frac{\delta \theta_7}{\delta \theta_5}$ are computed by first taking their partial differentials with respect to $\theta_1$ and $\theta_5$ (the active joint angles), and then formulating the linear algebraic equations for the system.

Finally, in Equation (8), the Jacobian of the double four-bar loops comes out to be

$$
J = \begin{bmatrix} \frac{L_1 Sin(\theta_1 - \theta_2)}{Sin(\theta_2 - \theta_4)} Sin\theta_4 & \frac{L_5 Sin(\theta_5 - \theta_6)}{Sin(\theta_6 - \theta_7)} Sin\theta_7 \\ -\frac{L_1 Sin(\theta_1 - \theta_2)}{Sin(\theta_2 - \theta_4)} Cos\theta_4 & -\frac{L_5 Sin(\theta_5 - \theta_6)}{Sin(\theta_6 - \theta_7)} Cos\theta_7 \end{bmatrix}.
\tag{8}
$$

Required torque vector is computed by inverting the Jacobian matrix ($J^T$) in the force ($F_c$) domain as

$$
\tau = J^T * F_c.
\tag{9}
$$

Here, the static force can be computed accurately through the anthropomorphic weight data of human upper-limb and exoskeleton weight itself.

Implementation

Section 3.2 is implemented for the demonstration of the Jacobian computation for a 2-loop exoskeleton in the algorithmic format. Assume a static force [10, 10] N is applied to the end-effector of the exoskeleton.

Step 1:    Initialize number of loops, $N = 2$.
Step 2:    NumberOfLinks, $3N + 1 = 7$.
Step 3:    Initialize LinkLengthValue, $L = [0.30\ 0.34\ 0.06\ 0.21\ 0.23\ 0.40\ 0.40]$.
Step 4:    LoopMatrix = AssignLinkNumber(2) = $\begin{bmatrix} 1 & 2 & 3 & 4 \\ 5 & 6 & 2 & 7 \end{bmatrix}$.

Step 5: ActiveAngle (dof) in each loop = AssignActiveAngleInLoops(2) = [1 5].
Step 6: Initialize ActiveAngleValue = [205 326].
Step 7: ListAllAngles = Anticlockwise direction from $X$-axis = [205 $\theta_2$ 0 $\theta_4$ 326 $\theta_6$ $\theta_7$].
Step 8: LoopClosureEquations = LoopClosureEq(2, LoopMatrix, LinkLengthValue, ListAllAngles) =

$$
\begin{aligned}
F_1 &= & L_2 cos(\theta_2) - L_4 cos(\theta_4) & & = 0.2808, \\
F_2 &= & L_2 sin(\theta_2) - L_4 sin(\theta_4) & & = 0.0707, \\
F_3 &= & L_6 cos(\theta_6) - L_2 cos(\theta_2) - L_7 cos(\theta_7) & & = -0.3925, \\
F_4 &= & L_6 sin(\theta_6) - L_2 sin(\theta_2) - L_7 sin(\theta_7) & & = -0.6469.
\end{aligned}
\tag{10}
$$

Step 9: InactiveAngles = LevenbergMarquardtAlgorithm(2, LoopClosureEquations, LinkLengthValue, ListAllAngles, ActiveAngle)
$\theta_2 = 13.80$, $\theta_4 = 106.19$, $\theta_6 = 39.77$ and $\theta_7 = 124.59$.
Step 10: AllAnglesValue = [205 336.46 285.41 278.03 326.10 272.82 255.20].
Step 11: JacobianNLoopFourBar = JacobianNLoopFourBar(2, LoopMatrix, LinkLengthValue, AllAnglesValue)

$$
J = \begin{bmatrix} 0.2599 & -0.5802 \\ 0.0366 & 0.1533 \end{bmatrix}.
$$

Step 12: Torque = $J^T * F$

$$
\tau = \begin{bmatrix} 2.96 \\ -4.26 \end{bmatrix}.
$$

*3.3. An Optimal Problem Formulation*

The optimization problem is formulated to generate an appropriate synthesis solution for the selected configuration to access task–space locations (TSLs) while improving kinematic performance. The Jacobian conditioning index and reachability at working locations are used as performance evaluation criteria and would be addressed using the genetic algorithm (GA) as the problem of finding acceptable configurations with good condition values. The conditioning index is a numerical value that indicates the dexterity of an end-effector stance. It denotes the Jacobian transformation's uniformity with regard to the direction of the joint rates, or local performance index. It is calculated using the Jacobian's eigenvalues. It is worth mentioning here that the selection of the performance index is just a matter of choice made in this work for representing the general platform for assisting a designer. Once the kinematic model and Jacobian are computed, any other performance index can also be worked upon. Given the computed Jacobian for a manipulator at a specific posture, the index (Condition Number, $C = \frac{\sigma_{max}}{\sigma_{min}}$ and Conditioning Index = $\frac{1}{C}$) can be computed using singular value decomposition (SVD) [26]. The $[0, 1]$ range of the conditioning index is commonly utilized. A higher score on the conditioning index indicates an improved kinematic performance. The lower conditioning index directs attention to the problem of a singularity. The manipulability fluctuates with minor posture changes. It does, however, require the highest manipulability rating on occasion. The formulated optimization problem for simulating natural human movement is defined below.

**Objective function**:
*Minimize*
Jacobian Condition Number = $(\sigma_{min}/\sigma_{max})$

**Subject to constraints**:
*End-effector Reachability*:
$(x_i - x_c)^2 + (y_i - y_c)^2 \leq \epsilon$
*Design limits*:
$L_{lower} \leq link\ length \leq L_{upper}$
$\theta_{lower} \leq joint\ angle \leq \theta_{upper}$
$-1 \leq x_s, y_s \leq 1$
$\theta_s = \text{Constant}$
*Kinematic constraints:*
$X_o = X_s, \& Y_o = Y_s;$
$X_P = X_W, \& Y_P = Y_W;$

where

$[x_i, y_i]$ = *i*th TSL position $\forall i \in [1, n]$,
$[x_c, y_c]$ = Current end-effector position,
$[x_s, y_s, \theta_s]$ = Shoulder position and orientation,
$[X_o, Y_o]$ = origin coordinates of exoskeleton,
$[X_P, Y_P]$ = end-point coordinates of exoskeleton,
$[X_s, Y_s]$ = origin coordinates of the human shoulder,
$[X_W, Y_W]$ = coordinates for wrist points.
$\sigma$ = Eigen value
$n$ = Number of TSLs
$m$ = Number of joints
$L$ = Link length
$\theta$ = Joint angle
$\epsilon$ = Tolerance limit

### 3.4. Performance Evaluation: Dimensional Synthesis

Dimensional synthesis was done in order to obtain the best conditioning index of the Jacobian matrix. Lifting the right hand is considered the task and the task space locations (human trajectory points) as a set of point $P$.

$$P = P_i, \forall i = 1\ to\ n \tag{11}$$

### 4. Results and Discussion

To demonstrate the proposed task-based dimensional synthesis algorithm, MATLAB R2015a is run on an Intel(R)Xeon(R)CPU E5-1607 v2 @ 3.00 GHz 3.00 GHz CPU equipped with 12 GB RAM. The average time to compute the results is 20 h. The formulated problem in Section 3.3 provides a method for synthesizing the double four-bar configuration optimally for the given task. To demonstrate the utility of the problem formulation, the wrist locations related to the shoulder as a reference frame, as given in Table 1, must be traced through the configuration.

### 4.1. Case-I: Towards Minimizing Conditioning Index Only

For initial analysis, using anthropomorphic human upper-limb data, the lower and upper bounds for link lengths are set to 0.02 m and 0.40 m, respectively, while joint angles are set to 0.01° and 360°, respectively. The formulated problem with double four-bar configuration connected in series represented the mechanism with link lengths $L_1, L_2, \ldots,$ $L_7$. However, lifting the right hand is considered the task and the task-space locations (TSLs) are represented as $P_1, P_2, \ldots, P_5$ (refer to Table 1). The optimal link lengths are obtained as 0.29, 0.33, 0.10, 0.20, 0.22, 0.39 and 0.40 m, respectively. The results that were accomplished while reducing the conditioning index, manipulability, and torque for both active angles corresponding to the reachability at each TSL are illustrated in Table 2. This table focuses on the double-four-bar arrangement that is connected in series. The range of the conditioning

index is between 0.03 and 0.60, the range of manipulability is 32.72 to 89.92 for the specified TSLs. The best postures are evaluated from $P_1$ to $P_4$ TSLs. Table 2 also includes the rated torque values required for both active actuators of the mechanism, which are determined to be between $-4$ N-m and $-1.5$ N-m for actuator-1 and between $-8.3$ N-m and 5.1 N-m for actuator-2. A negative toque implies rotation in the clockwise direction.

**Table 2.** Case 1: Optimal results obtained through emulating natural human motion.

| TSLs | Conditioning Index | Manipulability | Torque $\theta_1$ (N-m) | Torque $\theta_5$ (N-m) |
|------|--------------------|----------------|-------------------------|-------------------------|
| $P_1$ | 0.12 | 74.8 | $-2.9$ | 5.1 |
| $P_2$ | 0.42 | 32.72 | $-3.2$ | $-8.3$ |
| $P_3$ | 0.23 | 89.92 | $-3.4$ | 2.1 |
| $P_4$ | 0.41 | 28.06 | $-4.0$ | 3.6 |
| $P_5$ | 0.06 | 80.78 | $-1.5$ | $-8.1$ |

*4.2. Case-II: Toward Ergonomically and Aesthetically Compatible with Human-Limb Modified Design Limits and Introduce Joint Angle Continuity*

The findings of case I are further worked upon in terms of making the results more compatible and ergonomically and aesthetically pleasing. This is accomplished by making certain modifications to the design restrictions and adding a new target aimed at joint angle continuity. The modified design constraints based on anthropomorphic data specify lower and upper limit restrictions for connection lengths, as shown in Table 3. Both rows indicate the minimum and maximum values for each connection length from $L_1$ to $L_7$.

**Table 3.** Case 2: Modified link length limits. All dimensions are in meters.

|  | $L_1$ | $L_2$ | $L_3$ | $L_4$ | $L_5$ | $L_6$ | $L_7$ |
|--|-------|-------|-------|-------|-------|-------|-------|
| Lower bound | 0.02 | 0.06 | 0.02 | 0.02 | 0.10 | 0.17 | 0.02 |
| Upper bound | 0.40 | 0.15 | 0.06 | 0.40 | 0.20 | 0.25 | 0.40 |

The joint angles are set as 0.01 and 360 degrees, respectively. Furthermore, in this iteration, the continuity of the joint angles is included as another objective as well as being shown in Equation (12).

**Objective 2:**

$$Joint\ angle\ movement\ continuity = \sum_{i=1,j=1}^{n,m} \left( {}^{i+1}\theta_j - {}^{i}\theta_j \right) \tag{12}$$

Thus, the nature of the problem is modified, and mutiGA is applied. The revised optimal link lengths are 0.31, 0.08, 0.05, 0.28, 0.19, 0.24 and 0.37 m, respectively. Table 4 shows the results obtained through the minimizing of joint angle movement. It is obtained that the modified problem synthesized the configuration with improved Jacobian performance.

**Table 4.** Case 2: Optimal results obtained through minimizing joint angle movement.

| TSLs | Conditioning Index | Manipulability | Torque $\theta_1$ (N-m) | Torque $\theta_5$ (N-m) |
|------|--------------------|----------------|-------------------------|-------------------------|
| $P_1$ | 0.3 | 83.52 | $-0.4$ | 9.8 |
| $P_1$ | 0.39 | 39.58 | $-4.7$ | $-1.0$ |
| $P_3$ | 0.6 | 36.26 | $-4.4$ | 6.3 |
| $P_4$ | 0.57 | 34.85 | $-3.6$ | 1.7 |
| $P_5$ | 0.26 | 71.24 | 3.1 | $-1.1$ |

### 4.3. Case-III: Introduce Elbow Mapping Condition

The condition, elbow mapping, denotes the mapping of the mechanism's coupler position to the human elbow position. The corresponding condition's mathematical equation is as follows:

Elbow mapping $(x_k - x_e)^2 + (y_k - y_e)^2 \leq \epsilon$.

Here, $[x_k, y_k]$ represents $k^{th}$ coupler position, $\forall k = 1$ to $n$ and $[x_e, y_e]$ represents elbow position.

In this study, the obtained optimal link lengths are 0.25, 0.15, 0.04, 0.27, 0.19, 0.19 and 0.25 m, respectively. Table 5 displays the results acquired in the process of introducing the elbow mapping condition. However, it is noticed that the Jacobian conditioning indices acquired for each TSL are not as excellent as those in case II, but the misalignment problem is mitigated in case III.

**Table 5.** Case 3: Optimal results obtained through minimizing joint angle movement.

| TSLs | Conditioning Index | Manipulability | Torque $\theta_1$ (N-m) | Torque $\theta_5$ (N-m) |
|------|--------------------|----------------|-------------------------|-------------------------|
| $P_1$ | 0.26 | 37.89 | −3.9 | 1.1 |
| $P_2$ | 0.366 | 36.57 | −3.8 | −0.37 |
| $P_3$ | 0.27 | 37.51 | −3.5 | -0.65 |
| $P_4$ | 0.5 | 33.69 | −3.9 | 1.0 |
| $P_5$ | 0.36 | 29.91 | 0.62 | 1.5 |

### 4.4. Comparison

A comparison analysis of all three instances has been shown in Table 6, and it has been discovered that case II has a remarkably higher kinematic conditioning index and manipulability. Despite the fact that case III has the greatest minimum and maximum values of the conditioning index and manipulability, it also has the most variation and variance, and the lowest reachability. As a result, case II is shown to be more appropriate for the given task, as well as having superior kinematic performance and repeatability.

**Table 6.** Optimal results of all three cases of double-four bar configuration

| | Condioning Index | | Manipulability | | Reachability |
|---|---|---|---|---|---|
| | **Min** | **Max** | **Min** | **Max** | |
| **Case 1** | 0.0007944 | 0.2663 | 8.528 | 245.1 | 21.5405 |
| **Case 2** | 0.007461 | 0.5263 | 17.3 | 327 | 0.1612 |
| **Case 3** | $2.991 \times 10^{-5}$ | 0.9355 | 18.96 | $7.569 \times 10^9$ | 249.1004 |

## 5. Validation

MATLAB software is used to plot the instantaneous center of obtained optimal double four-bar configuration and the elbow positions. Figure 6 shows the dimensions obtained by varying the instantaneous center of a double four-bar configuration, approximately −15 cm to 10 cm in the $Y$-direction, and −18 cm to 10 cm in the $X$-direction during task completion, are graphically matched with the cross-sectional area (25 cm × 28 cm) due to change in the elbow end positions, approximately −39 cm to −11 cm in the $Y$-direction and 5 cm to 30 cm in the $X$-direction. This represents the closeness of changing patterns. The MATLAB plot validates that the area involved during motion between double four-bar configuration's instantaneous center with elbow positions are identical, which lies under the reported value of the cross-sectional area of the instantaneous center of elbow joint at the lateral view (sagittal plane) [23,24].

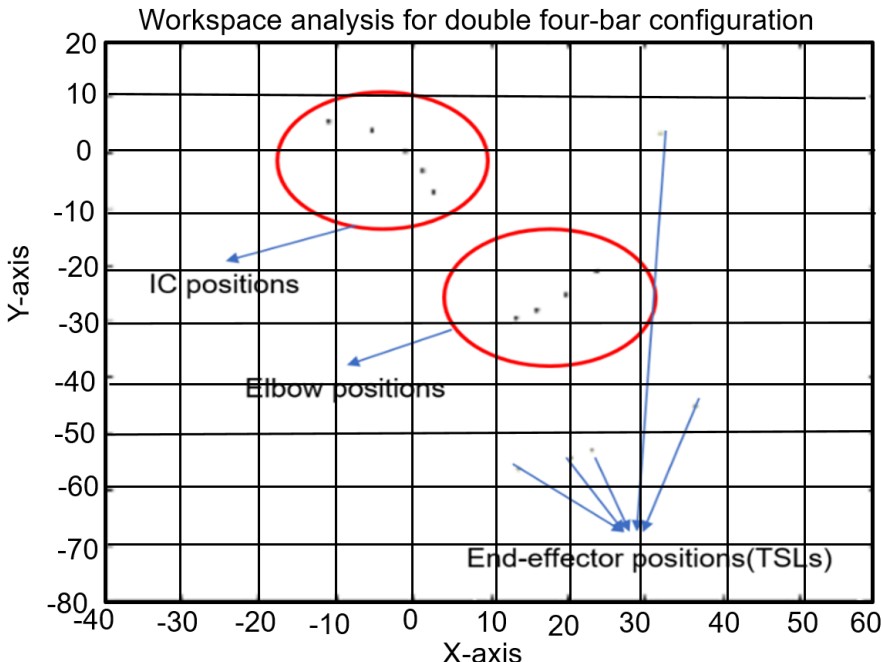

**Figure 6.** Pictorial view of area occupied by elbow positions, and the variation of the instantaneous center of double four-bar configuration during task completion.

The prototype of the designed configuration coupled with human limb is fabricated as shown in Figure 7, which shows the task performance while staying comfortable in movement, i.e., with least misalignment.

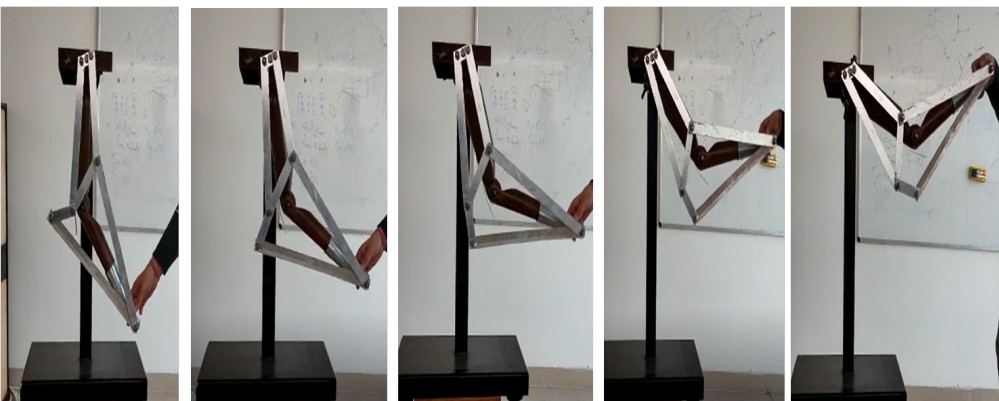

**Figure 7.** Prototype of coupled system.

Simulink's MATLAB toolbox is used to calculate the force experience at the human wrist as shown in Figure 8. The 3*D* simscape model and graph are shown in Figure 9 to reflect the force acting at the end-effector of the wrist. Force is measured in dynes (CGS system). An *X-Y-Z* force diagram is shown in the graph with blue lines, yellow lines, and an orange line, respectively. The constraint force lies in between $0.09 \pm 0.13$ N in the *X*-direction, $-0.001 \pm 0.004$ N in the *Y*-direction, and 0 N in the *Z*-direction. Therefore, the obtained wrist mobility force is within the acceptable tolerable force range (1 N).

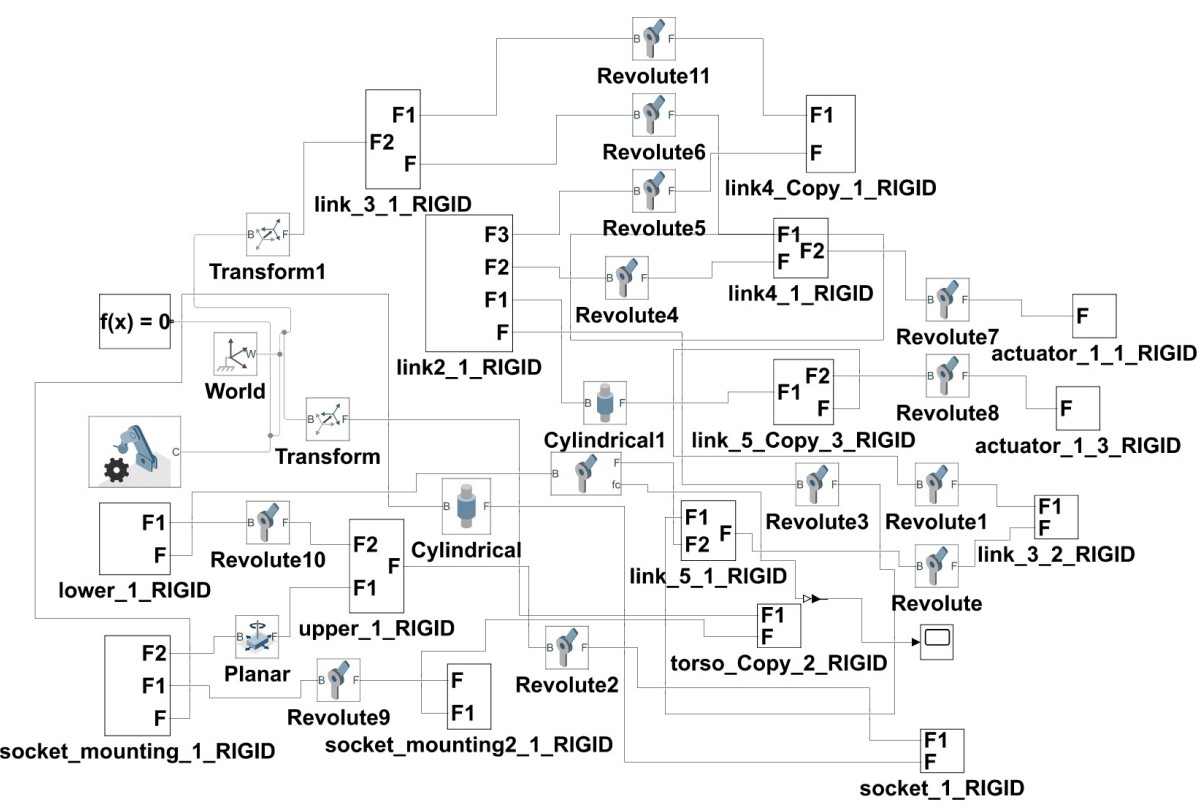

**Figure 8.** MATLAB simulink model of coupled system.

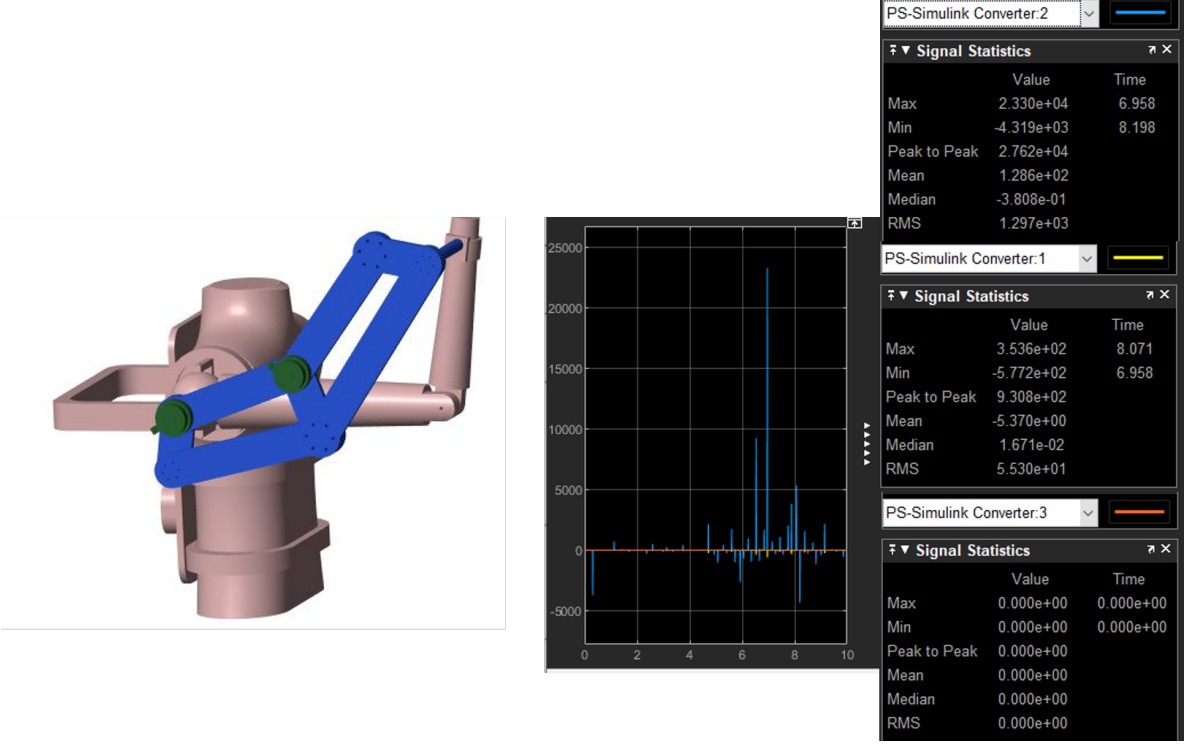

**Figure 9.** Simscape 3D rendering of the modelled exoskeleton.

## 6. Conclusions

The purpose of this research is to discuss the development of a rehabilitative exercise-based hybrid exoskeleton that avoids misalignment difficulties while replicating natural human mobility. This is achieved by the use of a double four-bar configuration. The kinematic modeling of this double four-bar system is formulated, which is further used for the optimal dimensional synthesis. Three optimal problem formulations are detailed to develop an exoskeleton for individuals that has good kinematic performance, and is ergonomically sound and visually appealing. The major aspects are design limitations, joint angle continuity, and emulating natural human motion. No acceptable solution is found in case III, which involves aligning the elbow joint's motion, whereas case II is adequate for the desired work and under acceptable conditions. When the instantaneous center of the four-bar design is measured, it is compared with normal human elbow locations in order to verify the results obtained. Both regions are found to be identical. Finally, the constraint force felt at the wrist is estimated with the MATLAB Simulink software, and the wrist's mobility comfort with an acceptable force, owing to coupling, is proven.

**Author Contributions:** Writing—original draft preparation, S.G.; Writing—review and editing, E.S. and A.A. All authors have read and agreed to the published version of the manuscript.

**Funding:** This research received no external funding.

**Institutional Review Board Statement:** Not applicable.

**Informed Consent Statement:** Not applicable.

**Acknowledgments:** The authors sincerely acknowledge the grant from Department of Science and Technology (DST) and Global Innovation and Technology Alliance (GITA) for financial support of this work via Affordable Preventive and Assistive Technology for Healthcare (A-PATH) project under INDO—UK joint applied R & D programme (Newton Bhabha Scheme).

**Conflicts of Interest:** There is no conflict of interest.

## Abbreviations

The following abbreviations are used in this manuscript:

| | |
|---|---|
| TSLs | Task space locations |
| dof | Degrees-of-freedom |
| ADL | Activities of daily living |
| ROM | Range of motion |
| $P$ | Exoskeleton end-effector |
| FPS | Frames per second |
| $F$ | dof of multi-loop chain |
| $f_k$ | Known dof joint |
| $f_{uk}$ | Unknown dof joint |
| $d$ | Function of motion space in closed-loop |
| $l$ | Number of separate loop |
| ICs | Instantaneous centers |
| $d_x$ | Linear displacement |
| $d_y$ | Angular displacement |
| D-H | Denavit–Hartenberg |
| $t$ | Set time duration |
| $n$ | Number of steps |
| GUI | Graphical user interface |
| $n$ | Number of TSLs |
| $m$ | Number of joints |
| $L$ | Link length |
| $\theta$ | Joint angle |

| | |
|---|---|
| $C$ | Elbow point |
| ESLs | Elbow space locations |
| $J$ | Jacobian matrix |
| $F_c$ | Force |
| $\tau$ | Torque |
| GA | Genetic algorithm |
| $N$ | Number of loop |
| $\sigma$ | Eigen value |
| $\epsilon$ | Tolerance limit |
| $[x_i, y_i]$ | $i$th TSL position $\forall i \in [1, n]$, |
| $[x_c, y_c]$ | Current end-effector position, |
| $[x_s, y_s, \theta_s]$ | Shoulder position and orientation, |
| $[X_o, Y_o]$ | Origin coordinates of exoskeleton, |
| $[X_P, Y_P]$ | end-point coordinates of exoskeleton, |
| $[X_s, Y_s]$ | Origin coordinates of the human shoulder, |
| $[X_W, Y_W]$ | Coordinates for wrist points |
| $[x_k, y_k]$ | $k$th coupler position, $\forall k = 1$ to $n$ |
| $[x_e, y_e]$ | Elbow position |

## Appendix A. Upper-Body Motion Modeling

### Motion Tasks

1. Eating soup by spoon (spatial movement)—eatsoup;
2. Eating fruits by fork (spatial movement)—eatfruit;
3. Drinking a cup of tea (spatial movement)—drinktea;
4. Lifting empty hand (planar movement)—lifthandright, lifthandleft, lifthandboth;
5. Lifting 5 kg load (planar movement)—lift5kgright, lift5kgboth.

### Naming Convention

**Example:**

*drinktea_s2_25m_001*

*Task_subjectnumber_age* & *gender_takenumber*

**Table A1.** Subject details.

| Test_Subject | Code | Height (m) | Age | Gender |
|---|---|---|---|---|
| S2 | s2_25m | 1.89 | 25 | M |
| S3 | s3_33m | 1.86 | 33 | M |
| S4 | s4_20m | 1.83 | 20 | M |
| S5 | s5_20m | 1.9 | 20 | M |
| S8 | s8_25f | 1.6 | 25 | F |
| S9 | s9_23f | 1.65 | 23 | F |
| S13 | s13_30m | 1.8 | 31 | M |
| S14 | s14_26m | 1.83 | 26 | M |

**Table A2.** Coordinates of other markers for lifting right hand at the difference of 50 frames in takes.

| Frame | Time (Seconds) | Rshoulder(M1) Position | | | Rshoulderback(M2) Position | | | RshoulderTop(M3) Position | | | RUArm(M4) Position | | |
|---|---|---|---|---|---|---|---|---|---|---|---|---|---|
| | | X | Y | Z | X | Y | Z | X | Y | Z | X | Y | Z |
| 0 | 0 | 0.016777 | 1.394482 | 0.075386 | −0.12263 | 1.418374 | −0.03393 | −0.13417 | 1.521386 | 0.080486 | −0.1404 | 1.404109 | 0.057184 |
| 50 | 0.416667 | 0.017202 | 1.394657 | 0.077207 | −0.12222 | 1.418768 | −0.03204 | −0.1336 | 1.521729 | 0.08244 | −0.13997 | 1.404472 | 0.059076 |
| 100 | 0.833333 | 0.018304 | 1.39429 | 0.076802 | −0.1233 | 1.419311 | −0.02938 | −0.13086 | 1.523144 | 0.08462 | −0.13912 | 1.405819 | 0.062205 |
| 200 | 1.666667 | 0.007082 | 1.397153 | 0.062984 | −0.15754 | 1.429429 | 0.001237 | −0.10818 | 1.556936 | 0.072941 | −0.14153 | 1.443649 | 0.092683 |
| 250 | 2.083333 | 0.0039 | 1.397069 | 0.06302 | −0.16143 | 1.429533 | 0.003316 | −0.1107 | 1.557285 | 0.073609 | −0.14415 | 1.444224 | 0.094455 |
| 300 | 2.5 | 0.006394 | 1.396812 | 0.062799 | −0.15837 | 1.429426 | 0.001614 | −0.10814 | 1.557143 | 0.072331 | −0.14188 | 1.44412 | 0.0929 |
| 350 | 2.916667 | 0.012425 | 1.395459 | 0.060533 | −0.15025 | 1.424128 | −0.00779 | −0.11029 | 1.549401 | 0.073116 | −0.13941 | 1.433874 | 0.084995 |
| 400 | 3.333333 | 0.009797 | 1.394876 | 0.074598 | −0.1445 | 1.416453 | −0.01304 | −0.13839 | 1.52284 | 0.098675 | −0.14831 | 1.404974 | 0.0801 |
| 450 | 3.75 | 0.012551 | 1.394673 | 0.079421 | −0.1233 | 1.418039 | −0.03439 | −0.14105 | 1.518437 | 0.081546 | −0.14404 | 1.401562 | 0.05572 |
| 500 | 4.166667 | 0.016579 | 1.395156 | 0.07754 | −0.12152 | 1.41996 | −0.03321 | −0.13396 | 1.522588 | 0.081455 | −0.14035 | 1.405421 | 0.057644 |

**Table A2.** *Cont.*

| Frame | Time (Seconds) | RElbowOut(M5) Position | | | RFArm(M6) Position | | | RWristOut(M7) Position | | | RWristIn(M8) Position | | |
|---|---|---|---|---|---|---|---|---|---|---|---|---|---|
| | | X | Y | Z | X | Y | Z | X | Y | Z | X | Y | Z |
| 0 | 0 | −0.27066 | 1.10472 | 0.041488 | −0.20706 | 1.108236 | 0.085806 | −0.27747 | 0.878572 | 0.13239 | −0.23676 | 0.915316 | 0.192978 |
| 50 | 0.416667 | −0.27036 | 1.105139 | 0.043364 | −0.20674 | 1.108619 | 0.087654 | −0.27733 | 0.87901 | 0.134369 | −0.23667 | 0.915744 | 0.194994 |
| 100 | 0.833333 | −0.27961 | 1.110757 | 0.069253 | −0.2109 | 1.112903 | 0.105246 | −0.2808 | 0.884008 | 0.162566 | −0.22836 | 0.918732 | 0.214746 |
| 200 | 1.666667 | −0.2942 | 1.541312 | 0.364715 | −0.23117 | 1.498246 | 0.378666 | −0.30574 | 1.583332 | 0.638904 | −0.22791 | 1.563722 | 0.623561 |
| 250 | 2.083333 | −0.28508 | 1.561679 | 0.364988 | −0.22382 | 1.51632 | 0.379511 | −0.29148 | 1.619174 | 0.637643 | −0.21555 | 1.59368 | 0.621453 |
| 300 | 2.5 | −0.28494 | 1.56368 | 0.361388 | −0.22465 | 1.517323 | 0.37679 | −0.29242 | 1.621899 | 0.63382 | −0.21685 | 1.595037 | 0.61814 |
| 350 | 2.916667 | −0.31758 | 1.445792 | 0.358787 | −0.25048 | 1.407794 | 0.367462 | −0.32747 | 1.42886 | 0.629513 | −0.24831 | 1.421016 | 0.610802 |
| 400 | 3.333333 | −0.31065 | 1.140836 | 0.183659 | −0.2337 | 1.134605 | 0.191511 | −0.30828 | 0.924533 | 0.314775 | −0.22943 | 0.945457 | 0.319622 |
| 450 | 3.75 | −0.27223 | 1.101506 | 0.036208 | −0.20195 | 1.10278 | 0.069086 | −0.26322 | 0.864316 | 0.103066 | −0.20024 | 0.895523 | 0.144746 |
| 500 | 4.166667 | −0.27455 | 1.107387 | 0.061292 | −0.20357 | 1.109471 | 0.092577 | −0.27163 | 0.875619 | 0.143278 | −0.20699 | 0.906838 | 0.182324 |

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
