# Peer review of "Toward Avoiding Misalignment: Dimensional Synthesis of Task-Oriented Upper-Limb Hybrid Exoskeleton"

_robotics, doi:10.3390/robotics11040074_

Round 1

Reviewer 1 Report

In this manuscript, the author developed a planar 2-dof task-oriented upper-limb rehabilitation exoskeleton for the recovery of shoulder and elbow flexion/extension movement while preventing joint misalignment and enhancing user comfort. The work is interesting; however, the originality and research gap the author tried to fill in is not very clear. I believe a thorough and in-depth review in this specific field is required. The work can be considered after revision.

Author Response

The authors acknowledge the reviewer's observations and have revised the text to clarify our rationale for doing the research. The primary challenges addressed by this research are the assistance of ROM exercises with the minimum number of active degrees-of-freedom (dof), the use of a double-four bar mechanism to avoid misalignment and singularity issues, and the provision of an optimal solution. Section-1 has been reworked to reflect the difficulties discovered during the literature review and editing, which have been included here in red.

 “Several robotic treatment devices have been developed to help with upper-extremity rehabilitation [3-7]. Only a few of these devices have been marketed as a result of their limitations, as detailed below.

In recent studies [8-11], it has been recommended that rehabilitation programmes target specific muscles and ligaments with more intense and regulated activities.  Rather than recreating a whole human workspace, splitting the workspaces is the best option. Rehabilitation firms are increasingly using rehab devices for upper extremity recovery because they can execute a greater number of therapeutically helpful movements in a smaller area. Task-based studies have been employed by researchers to develop upper-limb robotic rehabilitation devices that concentrate largely on ADL, even though the range of motion (ROM) is the first step before gaining independence in ADL [12, 13]. Mismatched rotational axes, high power-to-weight ratio, kinematic compatibility difficulties, and non-repetitive inverse solution may all result from serial connections in ADL-based manipulators with several degrees of freedom (dof) [6,14,15]. On the other hand, the researchers seek medically relevant motions with greater manipulability and positional reachability. The field lacks the contributions in task-oriented design for synthesizing robotic assistance with the lowest possible active dof. Secondly, serially linked connections are usually used to achieve high manipulability, but parallel manipulators are used to achieve greater positional reachability [16-20]. This concept inspired the use of hybrid configurations in this work. The shifting instantaneous centre gives the flexibility required to address misalignment and kinematic compatibility. Thus, an adequate hybrid configuration for simulating natural human motion is required.

This research focuses on a novel strategy utilising a hybrid arrangement to construct a 2-dof task-based rehabilitation device for the recovery of shoulder and elbow flexion/extension movement while preventing joint misalignment and enhancing user comfort as well as avoiding a large number of active dof. This is accomplished by incorporating the characteristics of a double-four bar mechanism and by performing dimensional synthesis.

Major aspects addressed in this paper in order to synthesise the architecture for rehabilitation aid proposed are as follows:

  • Task-based synthesis is used to design a customised upper-limb rehabilitation device with a minimal number of active degrees of freedom capable of acquiring therapeutically desirable movements (ROM exercises).
  • Designing and evaluating optimal double four-bar configuration to mimic the natural human motion and minimise misalignment and singularity concerns is offered.”

Reviewer 2 Report

Journal: Robotics (ISSN 2218-6581)

Manuscript ID: robotics-1748396

Title: Towards Avoiding Misalignment: Dimensional Synthesis of Task-Oriented Upper-Limb Hybrid Exoskeleton

Authors: Sakshi Gupta, Anupam Agrawal, and Ekta Singla

Comments and Suggestions for Authors:

GENERAL REMARKS

The Authors present the kinematic synthesis of an upper-limb exoskeleton designed to follow the center of rotation drift of human joints during motion. In particular, they present the use of a mechanism composed of two four-bar linkages sharing a common rod. Although the mechanism presented may have interesting insights from both clinical and technological perspectives, the article is very fragmented to read and difficult to understand. Specifically, it does not appear clear what the background of this work is, what is the actual contribution (Is it the mechanism itself? Is it the process – which, however, appears quite standard? Is it the idea behind the presented work?), and, often, different steps of the work are not properly linked together. Besides, the main text makes large use of symbols or concepts never defined in the manuscript, and syntax and punctuation need a thorough revision. For all these reasons, it is the reviewer’s opinion that the manuscript is not ready for publication and the Authors are encouraged to deeply revise it.

SPECIFIC COMMENTS

LINE 23 – it is reported: “Exoskeletons are capable of working on different upper-limb postures”. However, exoskeletons can work not only on the upper limbs. There seems to be a missing linking sentence here that takes the reader from the general case (i.e., exoskeletons for rehabilitation) to the particular case (i.e., exoskeletons for the upper limb).

LINE 45 – it is reported: “This is the basis for the use of hybrid configurations in this work.” In the previous text, it is not clear what exactly the Authors mean by “hybrid” nor why a hybrid solution is preferable. As the adjective “hybrid” is largely used with exoskeletons to indicate many different things (e.g., materials, actuation, etc.), the Authors should be more explicit on these two points.

FIGURE 2 – Figure 2(c) needs to be a standalone figure to better highlight what is the actual problem the manuscript is willing to address.

EQUATION 3 – can the Authors clarify the origin of all the numbers reported here?

LINE 125 – it is reported: “accessible anthropometric exercise-based data in rehabilitation centers.” Accessed databases and the corresponding rehabilitation centers should be stated in the text.

LINE 129 – this last sentence appears to be very ambiguous, please consider rephrasing it to better deliver its content to the riders.

LINE 138 – please add a reference to Open-Sim.

LINE 156 and TABLE 1 – in this form, the content of Table 1 is uninformative. Why did the Authors want to include the table in the article? The reviewer suggests adding a few lines of explanation to what is contained in the table.

SUBSECTION 3.2 – this whole section needs a proper revision. Please explain what the equations refer to and clearly define each of the symbols used.

SECTION 4 – Without an appropriate description of the parameters involved in the analysis, this section is poorly understandable.

TABLE 2 and TABLE 3 – please define all the quantities contained in the said tables.

LINE 273 and FIGURE 6 – it is reported: “It is demonstrated in Figure 6 that the obtained dimensions by the end-effector position i.e. -40 cm to -15 cm in y-direction, 5 cm to 32 cm in x-direction is matched graphically with the cross-sectional area due to change in the instantaneous center of a double four-bar configuration i.e. -15 cm to 10 cm in y-direction, -18 cm to 10 cm in x-direction during task completion.” What is given in the text does not seem to match what is shown in the figure. Please clarify both the figure and the text. Besides, Figure 6 is hardly readable without a proper grid.

FIGURE 8 – this figure is not much informative. Graphs showing the calculated force would fit better in the paper. Besides, the reviewer would suggest also adding an image of the Simscape 3D rendering of the modeled exoskeleton.

CONCLUSION – unfortunately, this section is not as strong as it should be with all the crucial details missing in the previous ones. The Authors are encouraged to revise it in light of all the changes suggested for the whole text.

Please, consider all these notes as a sincere interest in your work with the only aim of improving it.

Yours faithfully,

The reviewer

Author Response

The Authors present the kinematic synthesis of an upper-limb exoskeleton designed to follow the center of rotation drift of human joints during motion. In particular, they present the use of a mechanism composed of two four-bar linkages sharing a common rod. Although the mechanism presented may have interesting insights from both clinical and technological perspectives, the article is very fragmented to read and difficult to understand. Specifically, it does not appear clear what the background of this work is, what is the actual contribution (Is it the mechanism itself? Is it the process – which, however, appears quite standard? Is it the idea behind the presented work?), and, often, different steps of the work are not properly linked together.

Response: The authors acknowledge the reviewer's observations and have revised the text to clarify our rationale for doing the research. The primary challenges addressed by this research are the assistance of ROM exercises with the minimum number of active degrees-of-freedom (dof), the use of a double-four bar mechanism to avoid misalignment and singularity issues, and the provision of an optimal solution. Section-1 has been reworked to reflect the difficulties discovered during the literature review and editing, which have been included here in red.

 “Several robotic treatment devices have been developed to help with upper-extremity rehabilitation [3-7]. Only a few of these devices have been marketed as a result of their limitations, as detailed below.

In recent studies [8-11], it has been recommended that rehabilitation programmes target specific muscles and ligaments with more intense and regulated activities.  Rather than recreating a whole human workspace, splitting the workspaces is the best option. Rehabilitation firms are increasingly using rehab devices for upper extremity recovery because they can execute a greater number of therapeutically helpful movements in a smaller area. Task-based studies have been employed by researchers to develop upper-limb robotic rehabilitation devices that concentrate largely on ADL, even though the range of motion (ROM) is the first step before gaining independence in ADL [12, 13]. Mismatched rotational axes, high power-to-weight ratio, kinematic compatibility difficulties, and non-repetitive inverse solution may all result from serial connections in ADL-based manipulators with several degrees of freedom (dof) [6,14,15]. On the other hand, the researchers seek medically relevant motions with greater manipulability and positional reachability. The field lacks the contributions in task-oriented design for synthesizing robotic assistance with the lowest possible active dof. Secondly, serially linked connections are usually used to achieve high manipulability, but parallel manipulators are used to achieve greater positional reachability [16-20]. This concept inspired the use of hybrid configurations in this work. The shifting instantaneous centre gives the flexibility required to address misalignment and kinematic compatibility. Thus, an adequate hybrid configuration for simulating natural human motion is required.

This research focuses on a novel strategy utilising a hybrid arrangement to construct a 2-dof task-based rehabilitation device for the recovery of shoulder and elbow flexion/extension movement while preventing joint misalignment and enhancing user comfort as well as avoiding a large number of active dof. This is accomplished by incorporating the characteristics of a double-four bar mechanism and by performing dimensional synthesis.

Major aspects addressed in this paper in order to synthesise the architecture for rehabilitation aid proposed are as follows:

  • Task-based synthesis is used to design a customised upper-limb rehabilitation device with a minimal number of active degrees of freedom capable of acquiring therapeutically desirable movements (ROM exercises).
  • Designing and evaluating optimal double four-bar configuration to mimic the natural human motion and minimise misalignment and singularity concerns is offered.”

 Besides, the main text makes large use of symbols or concepts never defined in the manuscript, and syntax and punctuation need a thorough revision.

Response: The suggested corrections have been implemented. The manuscript is proofread thoroughly to ensure that such errors do not occur.

SPECIFIC COMMENTS

Point 1: LINE 23 – it is reported: “Exoskeletons are capable of working on different upper-limb postures”. However, exoskeletons can work not only on the upper limbs. There seems to be a missing linking sentence here that takes the reader from the general case (i.e., exoskeletons for rehabilitation) to the particular case (i.e., exoskeletons for the upper limb).

Response 1: The commented line is now modified in the revised detailed version of the introduction section.

Point 2: LINE 45 – it is reported: “This is the basis for the use of hybrid configurations in this work.” In the previous text, it is not clear what exactly the Authors mean by “hybrid” nor why a hybrid solution is preferable. As the adjective “hybrid” is largely used with exoskeletons to indicate many different things (e.g., materials, actuation, etc.), the Authors should be more explicit on these two points.

Response 2: The content of section 1 has been updated by the authors so that it better reflects the hybrid configuration meaning and the necessity of hybrid configurations, as can be seen by the additional lines below.

“Serially linked connections are usually used to achieve high manipulability, but parallel manipulators are used to achieve greater positional reachability [16-20]. This concept inspired the use of hybrid configurations in this work. The shifting instantaneous centre gives the flexibility required to address misalignment and kinematic compatibility. Thus, an adequate hybrid configuration for simulating natural human motion is required.”

Point 3: FIGURE 2 – Figure 2(c) needs to be a standalone figure to better highlight what is the actual problem the manuscript is willing to address.

Response 3: The authors have revised the manuscript and made a stand-alone Figure 2c as Figure 2 in the edited manuscript.

Point 4: EQUATION 3 – can the Authors clarify the origin of all the numbers reported here?

Response 4: The suggested corrections have been incorporated, as can be seen by lines below:

“Thus, for 2-dof planar exoskeleton as shown in Fig 2, the closed-loop consists of dof of the multi-loop chain: F is 2, total known dof (active dof):  is 4, function of motion space: d is 3 and number of separate loops: l is 1. Therefore, total unknown dof (passive dof ):  is computed by Eq 2.”

Point 5: LINE 125 – it is reported: “accessible anthropometric exercise-based data in rehabilitation centers.” Accessed databases and the corresponding rehabilitation centers should be stated in the text.

Response 5: The suggested corrections have been implemented

Point 6: LINE 129 – this last sentence appears to be very ambiguous, please consider rephrasing it to better deliver its content to the riders.

Response 6: The suggested corrections have been implemented and modified the line as

“Other recommended planar exercises can be carried out in a similar manner.”

Point 7: LINE 138 – please add a reference to Open-Sim.

Response 7: The suggested corrections have been implemented.

Point 8: LINE 156 and TABLE 1 – in this form, the content of Table 1 is uninformative. Why did the Authors want to include the table in the article? The reviewer suggests adding a few lines of explanation to what is contained in the table.

Response 8: Table 1 displays the sample coordinate points obtained by the Opensim software, which are then utilised for dimension synthesis in section 4. For explanation, the following lines are now added in the revised manuscript in section 4:

“The formulated problem in section 3.2 provides a tool for selecting a configuration to optimally synthesize the double four-bar configuration for the given task. To demonstrate the utility of the problem formulation, the wrist locations related to the shoulder as a reference frame, as given in Table 1 must be traced through the configuration.”

Point 9: SUBSECTION 3.2 – this whole section needs a proper revision. Please explain what the equations refer to and clearly define each of the symbols used.

Response 9: The authors have thoroughly revised the content in Subsection 3.2. The revised version is given below.

“In this section, kinematic analysis of human-robot exoskeleton is defined for double-four bar connected in series. The configuration's reference frame is aligned to the X-axis and assuming its lengths named as    to . All the joint angles are considered in an anti-clockwise direction from the X-axis, marked with to The configuration has two active joints, and . All the other joints ,  and  are inactive joints and can be expressed in terms of  and .

The closed-loop equations of the system are given in Eq 4,

= Cos  + Cos  − Cos − Cos  = 0,                           (4)
= Sin  + Sin  − Sin − Sin  = 0,
= Cos  + Cos  − Cos  − Cos  = 0,
= Sin  + Sin  − Sin  − Sin  = 0.

Consider elbow point C as a set of elbow-space locations (ESLs), which is computed in Eq 5 as

Cx = Cos  + Cos                                                  (5)                                                        
Cy = Sin  + Sin

The location of the end-effector, considered at point P, is computed in Eq. 6, through the path OACP.

Px= Cos  + Cos  + Cos                                        (6)                                             
Py = Sin  + Sin + Sin

Eq. 7 illustrates the Jacobian obtained using these point locations of the double four-bar mechanism.”

 Point 10: SECTION 4 – Without an appropriate description of the parameters involved in the analysis, this section is poorly understandable.

Response 10: The authors have thoroughly revised the content in section 4.

“To demonstrate the proposed task-based dimensional synthesis algorithm, MATLAB R2015a is running on an Intel(R)Xeon(R)CPU E5-1607 v2 @ 3.00GHz 3.00GHz CPU equiped with 12 GB RAM. The average time to compute the results is 20 hours. The formulated problem in section 3.2 provides a method for synthesising the double four-bar configuration optimally for the given task. To demonstrate the utility of the problem formulation, the wrist locations related to the shoulder as a reference frame, as given in Table 1 must be traced through the configuration.

4.1. Case-I: Towards minimizing conditioning index only

For initial analsis, using anthropomorphic human upper-limb data, the lower and upper bounds for link lengths are set to 0.02 m and 0.40 m, respectively, while joint angles are set to 0.010 and 3600 respectively. The formulated problem with a double four-bar configuration connected in series represented the mechanism with link lengths L1, L2, . . . , L7. However, lifting right hand has been considered as the task and represented the task-space locations (TSLs) as P1, P2, . . , P5 (refer to Table 1). The optimal link lengths are obtained as 0.29, 0.33, 0.10, 0.20, 0.22, 0.39 and 0.40 m, respectively. The results that have been accomplished while reducing the conditioning index, manipulability, and torque for both active angles corresponding to the reachability at each TSL are illustrated in Table 2. This table focuses on the double-four-bar arrangement that is connected in series. The range of the conditioning index is between 0.03 and 0.60, range of manipulability is 32.72 to 89.92 for the specified TSLs. The best postures are evaluated from P1 to P4 TSLs. Table 2 also includes the rated torque values required for both active actuators of the mechanism, which are determined to be between -4 N-m and -1.5 N-m for actuator-1 and between -8.3 N-m and 5.1 N-m for actuator-2. A negative toque implies rotation in the clockwise direction.

4.2. Case-II: Towards ergonomically and aesthetically compatible with human-limb modified design limits and introduce joint angle continuity

The findings of the case I are further worked upon in terms of making the results more compatible, ergonomically and aesthetically pleasing. This is accomplished by making certain modifications to the design restrictions and adding a new target aimed at joint angle continuity. The modified design constraints based on anthropomorphic data specify lower and upper limit restrictions for connection lengths, as shown in Table 3. Both rows indicate the minimum and maximum values for each connection length from L1 to L7.”

Point 11: TABLE 2 and TABLE 3 – please define all the quantities contained in the said tables.

Response 11: The suggested corrections have been implemented and added content in section 4.1 for Table 2.

“The results that have been accomplished while reducing the conditioning index, manipulability, and torque for both active angles corresponding to the reachability at each TSL are illustrated in Table 2. This table focuses on the double-four-bar arrangement that is connected in series. The range of the conditioning index is between 0.03 and 0.60, range of manipulability is 32.72 to 89.92 for the specified TSLs. The best postures are evaluated from P1 to P4 TSLs. Table 2 also includes the rated torque values required for both active actuators of the mechanism, which are determined to be between -4 N-m and -1.5 N-m for actuator-1 and between -8.3 N-m and 5.1 N-m for actuator-2. A negative toque implies rotation in the clockwise direction.

.For table 3, a few lines are added in section 4.2.

“The modified design constraints based on anthropomorphic data specify lower and upper limit restrictions for connection lengths, as shown in Table 3. Both rows indicate the minimum and maximum values for each connection length from L1 to L7.”

Point 12: LINE 273 and FIGURE 6 – it is reported: “It is demonstrated in Figure 6 that the obtained dimensions by the end-effector position i.e. -40 cm to -15 cm in y-direction, 5 cm to 32 cm in x-direction is matched graphically with the cross-sectional area due to change in the instantaneous center of a double four-bar configuration i.e. -15 cm to 10 cm in y-direction, -18 cm to 10 cm in x-direction during task completion.” What is given in the text does not seem to match what is shown in the figure. Please clarify both the figure and the text. Besides, Figure 6 is hardly readable without a proper grid.

Response 12: The authors extend their gratitude to the reviewer for his or her insightful comments and have adjusted figure 6 to include a grid and the text accordingly.

“Figure 6 shows the dimensions obtained by varying the instantaneous centre of a double four-bar configuration, approx. -15 cm to 10 cm in the y-direction, and -18 cm to 10 cm in the x-direction during task completion, are graphically matched with the cross-sectional area (25* 28 cm) due to change in elbow positions, approx. -39 cm to -11 cm in the y-direction and 5 cm to 30 cm in the x-direction. This represents how the closeness of changing patterns.”

Point 13: FIGURE 8 – this figure is not much informative. Graphs showing the calculated force would fit better in the paper. Besides, the reviewer would suggest also adding an image of the Simscape 3D rendering of the modelled exoskeleton.

Response 13: 3D Simscape model and graph have been updated to reflect the suggested changes (see Figure.). Force is measured in dynes (CGS system). An x-y-z force diagram is shown in the graph with blue lines, yellow lines, and an orange line respectively.

Point 14: CONCLUSION – unfortunately, this section is not as strong as it should be with all the crucial details missing in the previous ones. The Authors are encouraged to revise it in light of all the changes suggested for the whole text.

Response 14: The authors express their thanks to the reviewer for his or her thorough observations and highlight the changes in the conclusion accordingly in red colour.

“The purpose of this research is to discuss the development of a rehabilitative exercise-based hybrid exoskeleton that avoids misalignment difficulties while replicating natural human mobility. This is achieved by the use of a double four-bar configuration. The kinematic modelling of this double four-bar system is formulated which is used further the optimal dimensional synthesis. Three optimal problem formulations are detailed to develop an exoskeleton for individuals that has good kinematic performance, ergonomically sound and visually appealing. Majorly aspects are design limitations, joint angle continuity, and emulating natural human motion. No acceptable solution is found in Case III, which involves aligning the elbow joint's motion, whereas Case II is adequate for the desired work and under acceptable conditions. When the instantaneous centre of the four-bar design is measured, it has been compared with normal human elbow locations in order to verify the results obtained. Both regions have been found to be identical. Finally, the constraint force felt at the wrist is estimated with the MATLAB Simulink software, and the wrist's mobility comfort with an acceptable force owing to coupling is proven.”

Reviewer 3 Report

The paper treats the problem of exoskeletons for rehabilitation for neurological and muscular disorder patients

I have the following remarks:

The paper is interesting and addresses an important issue for the exoskeleton systems of patients with muscle and neurological impairments.

The theoretical results are confirmed by simulation but I think that a supplementation of the paper with experimental results, on an experimental model, would be useful.

An extensive comparison of the proposed method with results from literature papers would be useful.

Author Response

The authors are currently engaged in experimental work that is being conducted as a part of one of the projects and is not a part of this publication. To show the better clarity of findings, now the authors have added a 3D Simscape model and graph to reflect the confirmation of theoretical results (see Figure 9.). Force is measured in dynes (CGS system). An x-y-z force diagram is shown in the graph with blue lines, yellow lines, and an orange line respectively. The manuscript has been updated to reflect the modifications, which have been highlighted in the coloured text in the revised version of the manuscript.

Round 2

Reviewer 2 Report

The reviewer thanks the Authors for their work.